# *NR3C1* rs6198 Variant May Be Involved in the Relationship of Graves’ Disease with Stressful Events

**DOI:** 10.3390/biomedicines11041155

**Published:** 2023-04-12

**Authors:** Matheus Nascimento, Elisângela Souza Teixeira, Izabela Fernanda Dal’ Bó, Karina Colombera Peres, Larissa Teodoro Rabi, Adriano Namo Cury, Natália Amaral Cançado, Ana Beatriz Pinotti Pedro Miklos, Fernando Schwengber, Natássia Elena Bufalo, Laura Sterian Ward

**Affiliations:** 1Laboratory of Cancer Molecular Genetics, School of Medical Sciences, University of Campinas (UNICAMP), Campinas 13083-888, SP, Brazil; 2Department of Biomedicine, Nossa Senhora do Patrocínio University Center (CEUNSP), Itu 13300-200, SP, Brazil; 3Institute of Health Sciences, Paulista University (UNIP), Campinas 13043-900, SP, Brazil; 4Unit of Endocrinology and Metabolism, Santa Casa de Misericórdia de São Paulo, São Paulo 01221-010, SP, Brazil; 5Discipline of Endocrinology, School of Medical Sciences of Santa Casa de São Paulo (FCMSC-SP), Sao Paulo 01221-010, SP, Brazil; 6Endocrinology and Metabology Service of the Institute of Medical Assistance to State Civil Servants (IAMSPE), São Paulo 04029-000, SP, Brazil; 7Department of Medicine, Max Planck University Center, Indaiatuba 13343-060, SP, Brazil; 8Department of Medicine, São Leopoldo Mandic and Research Center, Campinas 13045-755, SP, Brazil

**Keywords:** Graves’ disease, stress, psychological, pituitary–adrenal system, single nucleotide polymorphism

## Abstract

Although stressful events are known to trigger Graves’ disease (GD), the mechanisms involved in this process are not well understood. The *NR3C1* gene, encoding for the glucocorticoid receptor (GR), presents single nucleotide polymorphisms (SNPs) that are associated with stress-related diseases. To investigate the relationship between *NR3C1* SNPs, GD susceptibility, and clinical features, we studied 792 individuals, including 384 patients, among which 209 presented with Graves’ orbitopathy (GO), and 408 paired healthy controls. Stressful life events were evaluated in a subset of 59 patients and 66 controls using the IES-R self-report questionnaire. SNPs rs104893913, rs104893909, and rs104893911 appeared at low frequencies and presented similar profiles in patients and controls. However, variant forms of rs6198 were rarer in GD patients, suggesting a protective effect. Stressful events were more common in patients than controls, and were reported to have clearly occurred immediately before the onset of GD symptoms in 23 cases. However, no association was found between these events and rs6198 genotypes or GD/GO characteristics. We suggest that the *NR3C1* rs6198 polymorphism may be an important protective factor against GD, but its relationship with stressful events needs further investigation.

## 1. Introduction

Stressful events have long been recognized as important environmental triggers of several autoimmune diseases, and have an intimate relationship with Graves’ disease (GD) [1,2]. However, the mechanisms that underlie this interaction have yet to be elucidated. One of the possible pathways involved in this stress/autoimmunity relation is the hypothalamic–pituitary–adrenal (HPA) axis, the main neuroendocrine regulator of mammalian homeostasis [3]. Glucocorticoid hormones (GCs) can be rapidly synthesized and secreted by the adrenals in response to stress through HPA axis signaling [4]. One of the main genes related to the stress response is *NR3C1*, which encodes the glucocorticoid receptor (GR), an intracellular receptor expressed in most cells [5]. Several GR protein variants are generated from the *NR3C1* gene by alternative splicing; all of these isoforms have unique patterns of tissue distribution and transcriptional regulatory profiles [6]. Once activated by the GC ligand, the GR + GC complex translocates into the nucleus, where it interacts with transcriptional coactivators or repressors of the expression of many genes [5]. GCs can modulate inflammatory gene expression through several distinct mechanisms. Many single nucleotide polymorphisms (SNPs) of the *NR3C1* gene have already been found to be directly or indirectly related (through GC action or GC resistance) to different mental disorders, especially the stress-related ones and autoimmune diseases such as multiple sclerosis, systemic lupus erythematosus, rheumatoid arthritis, inflammatory bowel disease, and membranous nephropathy [7,8,9,10,11,12,13]. Recent bioinformatics studies have suggested that there are several polymorphisms in the *NR3C1* gene, as well as in other genes such as TNF, CYP3A5 and *FKBP5*, that have deleterious potential to cause GC resistance [14,15]. Some investigations have suggested a relationship between *NR3C1* SNPs rs6189/rs6190, rs41423247, and rs6198 and autoimmune diseases, but studies on GD are still lacking [2,7,8,11,16].

SNP analysis is a great way to investigate the genetic influence on disease, but it has some challenges that need to be overcome, including choosing which SNPs are worth studying. In fact, there can be thousands of SNPs in each gene, most of them with a low minor allele frequency (MAF), which requires the investigation of large populations to obtain solid and conclusive results. This is one of the reasons why the results of SNP studies are complex and often controversial or inconsistent. An interesting solution for SNP selection is the use of bioinformatics tools. These tools can perform large-scale in silico simulations, performing complex analyses of all catalogued SNPs of a given gene to determine whether they can affect the structure and/or function of the proteins they encode. Therefore, it becomes possible to predict which SNPs are most promising for use in more expensive studies, such as case–control investigations with real-time qPCR that can confirm the simulation results. Using a combination of these bioinformatics tools with backing from the literature, we selected some *NR3C1* SNPs that had been investigated in a case–control study which assessed their association with stressful events.

## 2. Materials and Methods

### 2.1. Patients and Controls

This case–control study was approved by the Research Ethics Committees of the School of Medical Sciences—University of Campinas (FCM–UNICAMP), Campinas (SP), Brazil, the Institute of Medical Assistance to State Civil Servants (HSPE-IAMSPE), São Paulo (SP), Brazil, and the Santa Casa de Misericordia of São Paulo, São Paulo (SP), Brazil.

A total of 384 GD patients (325 women and 59 men, 41 ± 13 years old) were included in this study. All patients had confirmed GD and clinical and laboratory evidence of thyrotoxicosis, with suppressed thyroid-stimulating hormone (TSH), elevated values of triiodothyronine (T3) and free thyroxine (FT4), increased 24-h radioiodine and technetium uptake values, with a homogeneous and diffuse tracer distribution and/or the positivity of antibodies against the TSH receptor (TRAb). They were routinely followed up with periodic clinical and ophthalmological exams, laboratory dosages, and imaging exams according to the treatment consensus of GD and GO [17]. Of the patients, 209 (54.4%) presented eye disease. All patients had euthyroidism by the time of blood collection and interview, including the patients submitted to radioiodine therapy or to total thyroidectomy using levothyroxine. All patients were under periodic medical follow-up: 182 of them were treated with methimazole, 172 with radioiodine therapy and 15 underwent total thyroidectomy.

A control group of 408 healthy individuals (337 women and 71 men, 41 ± 12 years old) was composed of blood donors from our institution’s blood center. All patients and controls were carefully examined, with the exclusion criteria being those who were using any medication that could interfere with tests results, were less than 18 years old, were pregnant, or had a previous history of thyroid disease other than GD. All controls were carefully examined, with special attention paid to evidence of thyroid disease, and individuals with a previous or suspected history of any autoimmune condition were excluded. Part of the control group included 308 individuals with normal thyroid function and normal serum TSH and FT4 levels, as well as negative TgAb and TPOAb values. These healthy individuals were recruited in a previous study [18]. Peripheral blood was collected from each of the participants for the DNA extraction procedure.

Fifty-nine GD patients and sixty-six healthy controls accepted to respond to the self-reported Impact of Event Scale—Revised (IES-R) questionnaire, which was adapted for the Brazilian population [19,20]. This questionnaire was used to assess and validate the presence or absence of stressful events reported by individuals, as well as the contemporaneity of these events with the onset of GD symptoms.

### 2.2. SNPs Selection and In Silico Analysis

A combination of bibliographic surveys and bioinformatics tools was utilized to select the most promising SNPs of *NR3C1*. These tools were support vector machines (SVMs), which are a type of software that uses machine learning methods to simulate the effects that amino acid (AA) exchanges can generate on the structure and function of proteins. As a basis for this prediction, these machines use the nucleotide sequence of the gene in FASTA format (text-based format that represents both nucleotide and peptide sequences).

We used the NCBI platform to obtain the FASTA protein sequence of *NR3C1* (glucocorticoid receptor) (CCQ43043.1). The SIFT tool (Sorting Intolerant from Tolerant) simulated the effects of AA changes caused by the variants and allowed us to classify the SNPs as potentially benign, neutral, or deleterious [21]. Next, these selected *NR3C1* SNPs were analyzed by another type of software, which provided a more refined analysis of the interactions caused by AA exchanges, viz. PredictSNP, which is a consensus classifier that allows access to several tools: PolyPhen-2, SNAP2, MAPP, PhD-SNP, and Panther [22,23,24,25,26,27]. In addition to PredictSNP, we also utilized I-Mutant 3.0, MUpro, and PROVEAN as complementary tools [28,29,30,31].

To verify the protein–protein interactions of NR3C1, we employed the STRING database [32,33].

Not all SNPs in dbSNP are valid; some may be due to sequencing errors and/or may be unique to individuals, and this would justify not localizing the sequence in some tools. Thus, to prioritize the selection of functional SNPs, we took into account linkage disequilibrium (LD), which is the nonrandom association of alleles at two or more loci in a population and is directly related to a population’s history of mutation and recombination [34]. There was no significant LD between the selected SNPs. Unfortunately, even SNPs found to be deleterious through simulations may be rare and require an impractical population to validate.

### 2.3. DNA Extraction and Identification of Genotypes

DNA was extracted from blood samples using a standard protocol that included red blood cells and leukocyte lysis, urea and SDS treatment, phenol–chloroform extraction, and ethanol precipitation. Next, we analyzed the purity and concentration of the DNA samples by spectrophotometry. All DNA samples were genotyped for *NR3C1* SNPs rs6198 (C___8951023_10), rs104893913 (C__33877375_10), rs104893909 (C__27541057_30), and rs104893911 (C__33877384_10) by Real-Time PCR TaqMan^®^ SNP Genotyping Assays with a 7500 Real-Time PCR System (Applied Biosystems, Waltham (MA), USA). Reactions were examined using the allelic discrimination Endpoint Analysis mode of the 7500 Fast System Sequence Detection Software (SDS), version 1.4 (Applied Biosystems).

### 2.4. IES-R Questionnaire

We employed the Brazilian version of the IES-R questionnaire, which proposes the assessment of subjective stress related to life events, to verify the relation between stressful life events and the onset of GD. We used a short, revised version of this scale, consisting of 22 items [19,20]. As one of the most cited self-reported instruments for the assessment of post-traumatic stress symptoms, IES-R psychometric properties have been evaluated in several studies over the last few years, with good results that allow for the comparison of findings obtained in a given culture with data presented in several published papers and adapted to many different contexts and languages [35,36,37,38]. Despite not being indicated as a diagnostic tool for post-traumatic stress disorder, IES-R is adequate for the assessment of intrusive events, avoiding a hyperstimulation of these situations [20]. Only the events prior to the onset of symptoms were taken into account.

### 2.5. Statistical Analysis

Statistical analysis was performed using SAS software (Statistical Analysis System) for Windows, v9.4. (SAS Institute Inc., Cary, NC, USA). HaploView v4.2 software (Broad Institute, Cambridge, MA, USA) was used to calculate the Hardy–Weinberg equilibrium (HWE) and DL between SNPs. We also used Prism v9.0.0 software (GraphPad, San Diego, CA, USA) to construct the graphs [39].

We evaluated the association between variables and groups with the Mann–Whitney test. The chi-square test (χ^2^) or Fisher’s exact test were used for categorical variables. To calculate the odds ratio (OR) of the associated factors, univariate and multivariate logistic regression were used, with stepwise selection criteria. The OR and the 95% confidence interval (CI) provided a measure of the strength of association. The association between clinical variables and the genotypes of cases and controls was evaluated using the χ^2^ or Fisher’s exact test. The evaluation of the genotype balance was performed by HWE. The significance level which we adopted was 5%. A sample size of 384 cases and 384 controls was required for a power of calculation of 80% in relation to the genotypic investigation.

## 3. Results

### 3.1. In Silico Analysis

The in silico analysis flowchart is shown in Figure 1. We selected 878 missense SNPs from the NCBI platform database from a list of all catalogued SNPs of the *NR3C1* gene. In total, 63 out of these 878 were identified as deleterious by the multi-tool analysis of PredictSNP, and 24 of them showed deleterious potential according to most of the tools added to PredictSNP. Only 8 out of the 24 deleterious SNPs were selected for further analysis using other software because they had bibliographic records, which was essential according to our selection criteria. These 8 SNPs were all identified as deleterious by PredictSNP’s multi-tool analysis, and presented results that varied between neutral and deleterious according to the complementary tools, I-Mutant v3.0, Mupro, and PROVEAN (Figure 1).

The impact on NR3C1 protein stability was evaluated by I-Mutant and Mupro. Of the 8 SNPs, I-Mutant identified 4 that decreased stability and another 4 with no predicted effect on protein structural stability. Mupro found all 8 SNPs to decrease stability.

The effect of the SNP on the biological function of the protein based on sequence homology was analyzed by PROVEAN. Five SNPs were considered deleterious and two were considered neutral. PROVEAN did not identify the sequence of 1 SNP.

Finally, 4 SNPs showed the best results in the complementary tools. However, two of them were in linkage disequilibrium, so one was selected (rs104893911) and the other was discarded (rs104893912). Following all of these requirements, we selected 3 SNPs (rs104893913, rs104893909, and rs104893911) as the most potentially harmful to the structure, stability, and protein function (Figure 1). Next, we analyzed the 3 SNPs selected through the PredictSNP2 tool, together with the SNP 3’UTR rs6198. This SNP was initially selected by a bibliographic survey, and was later added to the in silico analysis using only PredictSNP2 and its integrated tools, since the other tools used in this work are intended exclusively for the analysis of missense SNPs. SNPs rs6198, rs104893913, rs104893909, and rs104893911 were predicted to be deleterious by PredictSNP2 and all its tools.

### 3.2. Analysis of Patients–Controls

We found no LD in the investigated SNPs of *NR3C1* (rs6198, rs104893913, rs104893909, and rs104893911). All the genotypes were in Hardy–Weinberg equilibrium (HWE-*p* > 0.05), in both the case and control groups. A sample size of 384 cases and 408 controls was achieved, which was considered adequate for a power of calculation of 80%.

Our data on smoking habits corroborate the findings in the literature, indicating an increased risk for GD onset in smokers. A multivariate analysis, adjusted for age and sex, confirmed the influence of smoking on the risk of GD (OR = 3.508; 95% CI = 2.294–5.364; *p* ≤ 0.0001). Since cases and controls are matched, there were no statistical differences between cases and controls for sex (*p* = 0.4392) or age (*p* = 0.1510).

The rs104893913, rs104893909, and rs104893911 SNPs did not show any variation in genotype: all patients and controls presented the wild-type variant for the respective SNPs.

Polymorphic genotypes of rs6198 (TC and CC) were more frequent in control individuals (T/T = 70%; T/C + C/C = 30%) than in GD patients (T/T = 78%; T/C + C/C = 22%) (*p* = 0.0176). The presence of the *NR3C1* polymorphism rs6198 represented a protective factor, considerably decreasing the chance of GD (OR 0.6463; 95% CI: 0.4488–0.9218).

We provide the genotype distribution and allele frequency of patients and controls in Table 1. A multivariate analysis, adjusted for age, sex, and smoking habits, confirmed that the inheritance of the TT genotype rs6198 of *NR3C1* (OR = 2.593; 95% CI = 1.630–4.123; *p* < 0.0001) was an independent factor in susceptibility to GD (Table 1).

We found no relationship between demographical or clinical variables and the genotypes of the cases or controls. We also did not observe any relationship between genotypes and GO (Table 2).

### 3.3. IES-R Questionnaire Evaluation

Stressful events were identified in 69% (n = 41) of the patients who answered the questionnaire, and 23 of them reported that the event had an impact and occurred immediately before the onset of GD symptoms. For the questionnaire evaluation, a sample size of 59 cases and 66 controls was achieved, which was considered adequate for a power of calculation of 95%. Stressful events were identified in only 38% (n = 25) of the control subjects (*p* = 0.0005). There was no association between these events and the genetic profile of the patients (*p* = 0.3827).

We were not able to demonstrate any association between stressful events and genotypes or between stressful events and any of the clinical features or evolution characteristics of the GD patients.

## 4. Discussion

Stress has long been recognized as a potential trigger for GD. Researchers have attempted to understand the relationship between stress and GD through numerous studies, each with varying conclusions. One such study, carried out by Sonino et al. in 1996, investigated the connection between stress and GD in 70 patients with the disease, 58 of whom were women and 12 of whom were men. The patients were matched for age, sex, marital status, and social class with 70 control subjects [40]. The researchers utilized Paykel’s Interview for Recent Life Events questionnaire to determine the number of stressful events reported by the patients and the controls. The study found that patients with GD reported significantly more stressful events than the control group.

However, other studies, such as Gray and Hoffenberg’s, have yielded contradictory results, suggesting that the relationship between stress and GD may be more complex than previously thought [41]. Some studies have proposed that the illness itself may cause stress, rather than the other way around. This theory suggests that patients with GD may experience stress due to the physical and emotional symptoms of the disease, rather than external factors [40,41].

Furthermore, the use of questionnaires to measure stress levels in patients with GD has been criticized for potential biases. Patients may selectively remember certain events or distort their communication of events, leading to inaccuracies in reported stress levels [41]. To minimize these biases, the researchers in this study opted to use a self-report questionnaire that has been widely validated in various populations, including in Brazil.

The self-report questionnaire alleviates the individual’s nervousness when responding to the questionnaire, as there is no need to speak to an interviewer while answering, which reduces (but does not exclude) the possibility of selective memory or distortion of the magnitude of events. There is also a basis in the literature for such statements, as several other studies have obtained similar results [40,42,43,44].

Despite having great potential to impair the structure, stability, and function of the protein, the epidemiological validation of the SNPs rs104893913, rs1048939109, and rs104893911 was disappointing, as it resulted in a 100% homozygous wild-type genotype in our population. Their MAF was too low, according to the ALFA (Allele Frequency Aggregator) and 1000 Genome projects [43,44]. Although demonstrating interesting results and having a great chance to play an important role in the pathogenic mechanism of Graves’ disease, as demonstrated by our bioinformatics data, SNPs that are not frequent in the population cannot be further explored as biomarkers of susceptibility to disease.

On the other hand, the *NR3C1* variant rs6198, which is located in an untranslated region of the gene, in exon 9β, has been widely associated with the GC resistance phenotype in previous studies, and provided interesting results in our GD patients and controls [10,45]. This variant affects the stability of GRβ isoform mRNA, consequently increasing the protein expression of GRβ, since it acts as a natural negative inhibitor of GRα (the only functional isoform of GR) transactivation of GC-responsive genes [46]. GRβ can inhibit GRα transcriptional activity by different molecular mechanisms, such as the formation of inactive dimers, competition for GC response elements, and interference with the activity of coregulators [47,48,49]. In addition, Bamberger et al. suggested that GRβ may be an endogenous inhibitor of GC action and may be an important dominant negative regulator determining glucocorticoid sensitivity in target tissues [50].

The rs6198 SNP has already been studied in many diseases, including autoimmune diseases, but there is a lack of data on GD. A systematic review and meta-analysis by Herrera C. et al. [51] found a protective role for the minor G allele of rs41423247 in autoimmune diseases, especially among Caucasians (OR = 0.78; 95% CI: 0.65, 0.92; *p* = 0.004). However, the data were insufficient to support evidence that other SNPs, including rs6198, modulated the risk of autoimmune diseases [51]. Here, we demonstrated that rs6198 is a protective factor for GD.

The role of GCs as triggers for GD is still unclear, but our results provide some clues in that direction. The imbalance caused by GC-mediated immunosuppression may be one of the critical factors contributing to the initiation of the pathological process. GRβ may, in fact, protect immune cells from immunosuppressive stress-related effects of high GC levels by inhibiting GC ligand binding to GRα. According to Liberman et al., GRβ is expressed at very low levels in most tissues; however, abundant GRβ expression has also been described, especially in some inflammatory cells such as lymphocytes and macrophages [49]. The role of GRβ in regulating GC action may also be related to the difference between the groups shown in the questionnaire results, which presented a greater number of patients with GD who experienced traumatic events in their lives that culminated in the appearance of GD symptoms. While most of the control subjects had also experienced traumatic events of the same nature as the patient group, such as grief, violence, accidents, social conflicts, etc., they remained “healthy”. On the contrary, when individuals carrying the GRβ isoform experienced these events, they may have precipitated the development of GD symptoms. We may speculate that the absence of the rs6198 minor allele, by elevating the expression of GRβ, could be related to GD development. In fact, these individuals may be unprotected from the immunosuppressive effects that GCs may cause when circulating in high levels after stressful events.

## 5. Conclusions

Our data suggest a protective role for *NR3C1* rs6198 in GD susceptibility, but further studies in larger cohorts with different ethnic profiles are needed to investigate an association with stressful events. The combination of a genotypic profile with other risk factors, such as smoking and other genetic characteristics, can configure a risk profile for GD and offer interesting prophylactic possibilities.

## Figures and Tables

**Figure 1 biomedicines-11-01155-f001:**
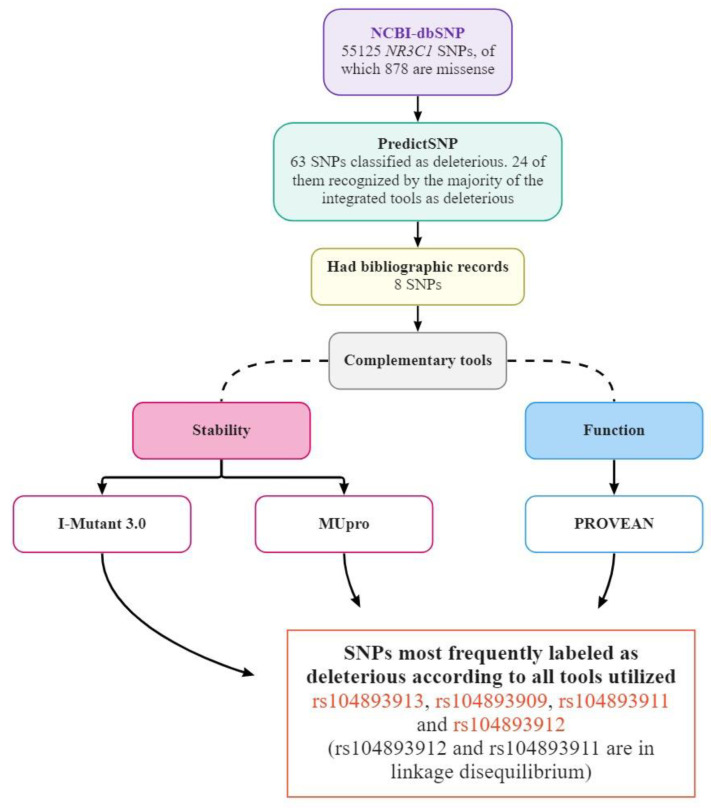
Flowchart of the sequence of in silico analysis of the SNPs of the *NR3C1* gene with tools analyzing the effects caused in the proteins by the exchanges of amino acids, induced by missense SNPs (elaborated upon by the authors).

**Table 1 biomedicines-11-01155-t001:** Multivariate analysis, adjusted for age, sex, and smoking habits, comparing SNPs of *NR3C1* (rs6198, rs104893913, rs104893909, and rs104893911) in 384 Graves’ disease patients and 408 controls.

SNP		Controls, N (%)	Cases, N (%)	*p* Value	OR	95% CI
***NR3C1* rs6198** Genotypes	TT	280 (68.6)	301 (78.4)	0.0038	1.627	1.170–2.264
	TC	120 (29.4)	80 (20.8)			
	CC	8 (2.0)	3 (0.6)			
Alleles	T	680 (83.3)	682 (88.8)	0.0017	1.586	1.187–2.128
	C	136 (16.7)	86 (11.2)			
***NR3C1* rs104893913** Genotypes	CC	408 (100)	384 (100)	N/A	N/A	N/A
	CT	0 (0)	0 (0)			
	TT	0 (0)	0 (0)			
Alleles	C T	816 (100) 0 (0)	768 (100) 0 (0)	N/A	N/A	N/A
***NR3C1* rs104893909** Genotypes	AA	408 (100)	384 (100)	N/A	N/A	N/A
	AT	0 (0)	0 (0)			
	TT	0 (0)	0 (0)			
Alleles	A T	816 (100) 0 (0)	768 (100) 0 (0)	N/A	N/A	N/A
***NR3C1* rs104893911** Genotypes	AA	408 (100)	384 (100)	N/A	N/A	N/A
	AG	0 (0)	0 (0)			
	GG	0 (0)	0 (0)			
Alleles	A G	816 (100) 0 (0)	768 (100) 0 (0)	N/A	N/A	N/A

SNP rs6198—TT = wild-type; TC = heterozygous; CC = polymorphic. SNP rs104893913—CC = wild-type; CT = heterozygous; TT = polymorphic. SNP rs104893909—AA = wild-type; AT = heterozygous; TT = polymorphic.

**Table 2 biomedicines-11-01155-t002:** Clinical features of 384 Graves’ disease patients with different genotypes of *NR3C1* rs6198.

Characteristics		rs6198		*p* Value
		TT	TC	
$\bar{x}$ ± SD	**Goiter Ultrasound (cm)**	27 ± 18	26 ± 18	-
	**Age**	42 ± 13	45 ± 13	-
**%**	**Sex**			0.8362
	Men	15.9	13.8	
	Women	84.1	86.3	
	**Smoking habit**			0.3546
	Yes	31.2	28.8	
	No	68.8	71.2	
	**TPOAb**			0.3329
	Positive	80.9	75.0	
	Negative	19.1	25.0	
	**TgAb**			0.2266
	Positive	52.3	46.1	
	Negative	47.7	53.9	
	**Ophthalmopathy**			0.4750
	Present	60.3	90.3	
	Absent	39.7	9.7	

TPOAb—thyroperoxidase antibody; TgAb—thyroglobulin antibody; TT—wild-type; TC—heterozygous. Note: For this analysis, the polymorphic genotype was disregarded, as there were only two observations.

## Data Availability

The data presented in this study are available upon request from the corresponding author. The data are not publicly available due to subjects’ privacy.

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
