# Peer review of "NR3C1 rs6198 Variant May Be Involved in the Relationship of Graves’ Disease with Stressful Events"

_biomedicines, 2023, doi:10.3390/biomedicines11041155_

Round 1

Reviewer 1 Report

The study by Matheus Nascimento et al. is focused on investigating possible mechanisms of the link between Graves’ disease and stressful events. To this end they examined four selected single nucleotide polymorphisms of NR3C1 gene that encodes glucocorticoid receptor and they found that the wild type of allele at rs6198 is related to an increased risk of the disease. The study is well designed and thorough. Although it has limited implications for clinical practice it is quite interesting as a basic research. From this point of view the study includes a methodologically interesting part on the identification of potentially involved SNPs - the use of bioinformatics tools. As the authors stated “these tools can perform large-scale in silico simulations, performing complex analyzes of all cataloged SNPs of a given gene to determine if they can affect the structure and/or function of the proteins they encode. Therefore, it becomes possible to predict which SNPs are most promising for use in more expensive studies” (lines 62-64). Then, they describe the multistage process of identifying those SNPs that led them to isolation of 3 promising SNPs. And yet those “more expensive studies” led to a huge disappointment: no polymorphisms were found. That is not shameful but it was concluded with one short sentence in the discussion. I’d expect a thorough discussion of this failure of  ‘in silico simulations’ that would be helpful for other researchers conducting studies on SNPs. Anyway, the discussion is the weakest part of the paper. It is too short and, what’s even more important, the line of reasoning is very unclear when the authors try to explain the possible connection between the rs6198 polymorphism and Graves’ disease. I guess that the authors believe the wild type of allele at rs6198 is related to the increased expression of GRβ isoform of glucocorticoid receptor that decreases cellular response to cortisol, but “the 3’UTR NR3C1 variant rs6198” is not clear for me. Then there is a sentence “The GRβ role in regulating GC action could also explain our questionary results, which showed a higher number of GD patients having experienced traumatic events in their lives in comparison to the control group, mostly just before the appearance of GD symptoms” (lines 284-286). I can’t understand how the of GRβ isoform could affect the probability of experiencing traumatic events in one’s life. The next statement: “In fact, these individuals may be unprotected from the immunosuppressive effects that GCs may cause when circulating in high levels after stressful events” also needs further explanation. It may be surprising for some readers that immunosuppression caused by GCs might trigger an autoimmune disorder.
Altogether, I would suggest rewriting and expanding the discussion before the publication of the study.

And one more minor remark:

 “Our data on smoking habits corroborates literature findings, indicating an increased risk for GD onset. A multivariate analysis adjusted for age, sex and smoking habits confirmed this finding (OR = 3.508; 95% CI = 2.294 – 5.364; p = < 0.0001)” (lines 200-201) – It is not clear what the given OR refers to. First sentence suggests it’s smoking but then why it was adjusted for smoking habits. There seems to be some contradiction.

Reviewer 2 Report

The Authors have studied the impact of some sequence variants in the NR3C1 gene susceptibility to Graves' disease and found that the rs6198 polymorphism was associated with a protective effects against the development of GD.

I have the following comments:

If the sample calculation was correct, how the Authors could explain that all the participants carried only the wt allele of most of polymorphisms investigated in this study?

I suggest to change the title as they have analysed only the impact of the rs6198 variant.  The in-silico analysis of other polymorphisms is redundant, since they are only showing the effect of the rs6198. In addition, similar results have been recently shown elsewhere (eg. https://doi.org/10.3390/biom12091307).

Minor: 

I suggest to change “NR3C1, a gene implicated in glucocorticoid resistance” as “NR3C1 gene encoding for the glucocorticoid receptor”

I suggest to better detail which investigations were performed in controls (thyroid US scan, blood testing) to exclude thyroid diseases.

Round 2

Reviewer 2 Report

I have some concern regarding how my comments have been addressed

1)       Genetic association studies are sufficiently powered only to test common genetic variants, because rare variants are found in a small number of individuals and need of huge samples. Thus, I think that it would be more correct to state in the Methods that the selected population was insufficient to study the effect of the rare variants obtained in the in-silico analysis.  In my opinion, the absence of these variants in this population is not a "result" of the study. The study was simply underpowered to analyse the effects of these variants.

2)       I still find redundant the in-silico analyses. Please consider to shorten this paragraph, since they have not analysed the effect of most of these variants and a similar study has been already performed.

Author Response

Once again, we value the care that the Editors and Reviewers are taking with our article. We took into account the remarks of the reviewer and, in response to his comments, amended the text accordingly.  The modifications are highlighted in yellow in the newly uploaded version of the manuscript and described below.

We hope that this lasted version of the manuscript will be accepted for publication in Biomedicines since this is an important subject and we believe that our manuscript brings important data to the field of Graves’s disease.

Response to reviewer

Thank you for your comments and suggestions, which were all accepted as detailed below.

  • “Genetic association studies are sufficiently powered only to test common genetic variants, because rare variants are found in a small number of individuals and need of huge samples. Thus, I think that it would be more correct to state in the Methods that the selected population was insufficient to study the effect of the rare variants obtained in the in-silico analysis. In my opinion, the absence of these variants in this population is not a "result" of the study. The study was simply underpowered to analyse the effects of these variants.”

Response: You are correct, our study was in fact underpowered to analyze the selected variants. We accepted your suggestion and added a paragraph concerning this issue in the last sentence of the Methods section:

“Unfortunately, even SNPs found to be deleterious through simulations may be rare and require an impractical population to validate.”

In addition, a sentence concerning this point was already in our Introduction – lines 61 to 64: “SNP analysis is a great way to investigate the genetic influence on disease, but it has some challenges that need to be overcome, including choosing which SNP is worth studying. In fact, there can be thousands of SNPs in each gene, most of them with a low minor allele frequency (MAF), which requires the investigation of large populations to obtain solid and conclusive results.”

Also, we have an entire paragraph discussing this point in the Discussion section, lines 277-284: “Despite having great potential to impair the structure, stability and function of the protein, the epidemiological validation of the SNPs rs104893913, rs1048939109 and rs104893911 was disappointing, as it resulted in a 100% homozygous wild-type genotype in our population. Their MAF was too low according to the ALFA (Allele Frequency Aggregator) and 1000 Genome projects [43,44]. Although demonstrating interesting results and having great chance to have important role in the pathogenic mechanism of Graves’ disease, as demonstrated by our bioinformatics data, SNPs that are not frequent in the population cannot be further explored as biomarkers of susceptibility to disease.”

  • “I still find redundant the in-silico analyses. Please consider to shorten this paragraph, since they have not analysed the effect of most of these variants and a similar study has been already performed.”

Response: We accepted your suggestion and shortened the paragraph.

Round 3

Reviewer 2 Report

I have no further comments